# Fibrotic Contracture of the Infraspinatus Muscle with or without Contracture of the Teres Minor Muscle: A Retrospective Study in Eight Dogs

**DOI:** 10.3390/ani14172589

**Published:** 2024-09-06

**Authors:** Androniki Krystalli, Sofianos Papaefthymiou, Kornilia Panteli, Aikaterini Sideri, Elena I. Pappa, Nikitas N. Prassinos

**Affiliations:** 1Surgery & Obstetrics Unit, Companion Animal Clinic, School of Veterinary Medicine, Faculty of Health Sciences, Aristotle University, 54627 Thessaloniki, Greece; sofpapae@windowslive.com (S.P.); korniliapanteli@hotmail.com (K.P.); ngreen@vet.auth.gr (N.N.P.); 2Clinic of Surgery, Faculty of Veterinary Science, School of Health Sciences, University of Thessaly, 43100 Karditsa, Greece; ksideri@vet.uth.gr (A.S.); elenpapp@vet.uth.gr (E.I.P.)

**Keywords:** contracture, dog, infraspinatus muscle, teres minor muscle, tenotomy

## Abstract

**Simple Summary:**

Muscle contracture refers to the pathologic process that results in fibrosis and permanent damage to a muscle. Theoretically, any muscle can be affected. This study retrospectively reviewed eight cases of the fibrotic contracture of the canine infraspinatus muscle. The treatment was the tendon’s tenotomy at its insertion and the release of all adhesions. In three cases, a teres minor simultaneous contracture was detected intraoperatively, which was treated via the tenotomy of the affected muscle. This is the first study to report the contracture of the teres minor muscle and its simultaneous coexistence with the fibrotic contracture of the infraspinatus muscle. Postoperatively, strict limitation of physical activity for two weeks followed by a gradual return to the full limb’s activity and analgesic and anti-inflammatory drugs were prescribed. The outcome was excellent, as the lameness resolved within two weeks in all cases. This retrospective study aims to compare its results with the literature to enrich the data on muscle contracture. Contracture of the infraspinatus and teres minor muscle should routinely be considered as a cause of thoracic limb lameness.

**Abstract:**

(1) Background: Fibrotic contracture of the canine infraspinatus muscle (FCIM) is considered an uncommon musculotendinous condition mainly affecting hunting dogs. After an acute onset of a painful non-weight-bearing lameness over a period of one to four weeks, a characteristic circumducted gait is developed in the affected thoracic limb. (2) Methods: Eight client-owned dogs of varying breeds, both sexes, aged 4–9 years old, and weighing 14–26 kg participated in the study. The duration of lameness prior to their first consultation ranged from 10 to 450 days. All participants were thoroughly examined clinically, orthopedically, and radiographically. (3) Results: The dogs underwent infraspinatus tenotomy, resulting in improved limb function. In three cases, a teres minor muscle contracture was revealed intraoperatively and resolved via a tenotomy at its insertion. The findings showed that 15 days post-operation, all dogs returned to full activity. The results obtained confirm and reinforce the literature data around FCIM and describe the first recorded condition of the simultaneous contracture of the infraspinatus and teres minor muscles. (4) Conclusions: In every case of shoulder lameness, contractures of all shoulder muscles should be included in the differential diagnosis and patients should be assessed for concurrent contractures even if infraspinatus contracture is identified.

## 1. Introduction

Dogs bear about 60% of their weight on their thoracic limbs [1]. The detection of a painful limb is accomplished by observing the animal’s gait, during which the head is lifted when the painful limb bears weight [2]. The shoulder joint is a diarthrodial joint consisting of the scapular glenoid and the humeral head [3]. Passive and active mechanisms are responsible for its stability and keep the joint restrained. The passive mechanisms consist of the glenohumeral ligaments, the joint capsule, the joint morphology, and the synovial fluid adhesion and cohesion, while the active mechanisms consist of the biceps brachii, subscapularis, supraspinatus, infraspinatus, and teres minor muscles and tendons [4]. The more important motions of the canine shoulder joint are flexion (47–57º ROM), extension (159–165º ROM), abduction (30º ROM), adduction, and internal and external rotation [5].

Fibrotic contracture of the infraspinatus muscle (FCIM) is an unusual cause of shoulder lameness [6,7,8]. It was first discovered in the Netherlands in 1970 and since then has been reported in various studies [9,10,11]. The tendon of the infraspinatus muscle passes over the joint capsule and inserts ventral to the cranial portion of the major tubercle of the humerus, functioning as a lateral collateral ligament. The infraspinatus bursa lies medial to the tendon and lateral to the joint capsule. The infraspinatus muscle can rotate the limb externally, as well as extend or flex the shoulder, depending on the position of the shoulder joint at the time of muscle contraction [3]. According to Blood and Studdert 2021 [12], contracture is defined as “the abnormal shortening of muscle tissue, rendering the muscle highly resistant to stretching”. Theoretically, contracture can affect any muscle, but it is most commonly reported on the infraspinatus and quadriceps and less commonly on the supraspinatus, teres minor, sartorius, gracilis, semitendinosus, and brachialis muscles [7,11,13,14,15,16]. FCIM usually involves medium-body-size working dogs or very active domestic dogs and, occasionally, cats [7,8,13]. It is typically unilateral; however, bilateral contracture also occurs [14,15].

Acute clinical signs include local pain, lameness, and swelling, which subside over one to four weeks after which the dogs progressively develop a persistent deformity, due to contracture [10,15]. In the standing position, they present elbow adduction and external rotation of the distal thoracic limb, and when walking, they show a characteristic circumducted gait abnormality of the distal limb with a carpal flip [6,11,17,18,19]. Atrophy of the spinatus muscle group and reduction in the range of shoulder flexion are frequently evident during orthopedic examination [8,13,20,21,22]. The causes of muscle contracture include repetitive strains and ischemia, infection, eosinophilic myositis, neoplasia, and myositis ossificans [14,15,23]. The most likely cause of FCIM appears to be trauma during exercise, leading to compartment syndrome, as hemorrhage, degeneration, atrophy, and fibrosis are the histological changes shown in infraspinatus muscle biopsies [8,20].

Radiographs of the shoulder joint appear normal, although, in some cases, they could show a narrowing of the scapulohumeral joint space [24], while ultrasonographic findings are a crucial diagnostic tool [8]. Biochemical blood analysis may show an elevation in CK [25]. The treatment consists of the tendon’s tenotomy at its insertion and the breakdown of adhesions [8,19]. The prognosis is good, as the animals return to their vigorous life in one or two months [10,26].

The teres minor tendon is smaller than the infraspinatus tendon, is located more caudally, inserts on the proximolateral aspect of the proximal humerus caudal and distal to the insertion of the infraspinatus tendon [27], and acts as a flexor of the shoulder joint. A myopathy of the teres minor muscle has also been reported as a cause of shoulder lameness. Ultrasonography was used for this diagnosis, and surgical resection of the affected tissue resolved the lameness [28].

The published information about infraspinatus contracture based on clinical cases is poor, while the simultaneous ipsilateral contracture of teres minor muscle is reported for the first time. The aim of the present study is to compare the published clinical cases, which illustrate these unusual soft tissue disorders, with our results and to present the clinical findings, surgical procedure, and follow-up of surgically treated dogs.

## 2. Materials and Methods

The study retrospectively examined the clinical records of eight client-owned dogs that presented to the Surgery and Obstetrics Unit at the Companion Animal Clinic, Department of Veterinary Medicine, Aristotle University of Thessaloniki, Greece, as well as at the Clinic of Surgery, University of Thessaly, Greece, who underwent tenotomy of the infraspinatus muscle between April 2004 and November 2018. Three of them also underwent tenotomy of the teres minor muscle. Each dog owner answered a dog-mobility questionnaire (Table 1) one year postoperatively to provide additional data about the patients.

The initial examination included the recording of the dog’s history (its nature, occurrence, and type of lameness (Figure 1), physical activity, presence of pain, and type and quantity of food), along with a physical and orthopedic examination and gait evaluation based on a six-grade scale (Table 2).

The findings were placed in the clinical records. Each affected joint was evaluated in order to detect the presence of pain during passive movement. A radiological examination using medio-lateral and ventro-dorsal views was obtained after physical examination, while the dogs were anesthetized.

The criteria for inclusion of animals in the study were as follows:Post-operative follow-up or communication with the owner for a period of ≥1 year from surgery;Good general health condition during surgery;Lameness of the affected limb due only to the fibrotic contracture of the infraspinatus muscle with or without contracture of the teres minor muscle. Any other cause was excluded (e.g., neurological disorder and osteoarthritis of the elbow or carpus joint);Grade of the non-affected limbs’ lameness of ≤2 degrees.

After induction of general anesthesia and aseptic preparation, a craniolateral approach of the scapulohumeral joint was performed in all cases. After a proximal displacement of the acromial part of the deltoid muscle, preparation of the infraspinatus muscle tendon, incision 1 cm close to its insertion to the greater tubercle (Figure 2) of the humerus, and release of adhesions to the shoulder joint capsule was obtained. Forced flexions of the shoulder and elbow joints were performed to ensure an unrestricted range of motion. In three cases, after subjecting the joint to passive movements, we realized that the contracture was not yet treated. Preparation of the teres minor muscle and partial or total incision to its insertion close to the humeral bone was performed (Figure 2). After re-subjecting the joint to passive movements, the condition was determined as treated.

The dogs were housed for 1 day in the clinic, where intravenous antimicrobial and analgesic drugs were given, and they were discharged with a prescription of carprofen (4 mg kg^−1^ bodyweight) (Rimadyl; Zoetis Hellas S.A, Athens, Greece) or firocoxib (5 mg kg^−1^ bodyweight) (Previcox, Boehringer Ingelheim Vetmedica GmbH, Ingelheim/Rhein Germany) orally twice a day for 5 to 10 days and amoxicillin–clavulanic acid (20 mg kg^−1^ bodyweight) (Synulox; Pfizer Hellas A.E., Athens, Greece) orally twice a day for 7 days. A significant limitation of physical activity (no running, no jumping, and only short walks with a leash) for 15 days was also advocated, and a re-examination 15 days postoperatively was recommended. When the examination findings were satisfactory, a gradual return to full activity, avoiding vigorous exercise for the next 15 days, was recommended. The postoperative clinical evaluation of the operated limb was assessed using the six-grade scale (Table 2).

Dog owners answered a questionnaire (Table 1) by phone, comprising eight questions related to compliance with the post-operative instructions, the patient’s current well-being and physical function, and the owner’s satisfaction with the outcome. The owners were asked about the method and duration of restraint of the dog and the efficacy of the analgesic. In addition, the limb’s time of final weight bearing and the grade of lameness after exercise were determined. Finally, the owners were asked to assess their dog’s quality of life, possible changes in its behavior, and the dog’s progress.

## 3. Results

From April 2004 to November 2018, the medical records of cases that had undergone tenotomy of the infraspinatus muscle were obtained from the registry of the two Companion Animal Clinics in Greece. Three of the patients underwent to tenotomy of the teres minor muscle too. After checking the patients’ compliance with the inclusion criteria, they were included in the present study.

The study group comprised eight dogs. Most of them were female (n = 5, 62.5%) and non-neutered (n = 6, 75%), with three male (37.5%) and two neutered (25%). Their age ranged from 4 to 9 years (mean = 6.25, median = 6) and their body weight from 14 to 26 kg (mean = 19.1, median = 18). There were varying breeds of dogs included in our study (Table 3); however, all of the dogs were used for hunting.

According to the case histories, the main mentioned symptom was an acute onset of shoulder lameness. Sudden lameness was the result of trauma in three cases (No = 1, No = 7, and No = 8) and of vigorous exercise in the rest of them. Preoperatively, the duration of lameness was 10 to 450 days (mean = 11.87, median = 60) before the dogs’ first presentation in the Veterinary Clinic. Five of them had already received medication with non-steroidal anti-inflammatory drugs (NSAIDs) for a period of 10–15 days, sometimes without improvement (No = 1, No = 2, and No = 5) and sometimes with a relapse (No = 3 and No = 8) after the end of the treatment. The preoperative gait evaluation showed grade 2 lameness in these dogs and grade 1 in the remainder (Table 2), characterized by a circumduction movement as the limb advanced during the stride and showing a flip-like movement of the paw when placing the limb. Their standing posture exhibited elbow adduction with external rotation of the distal part of the affected limb.

During an orthopedic examination, atrophy of the infraspinatus, supraspinatus, and spinous deltoideus muscles was observed in six cases (No = 1, No = 3, No = 4, No = 6, No = 7, and No = 8). When the dogs were examined in lateral recumbency, reduced shoulder flexion was evident in four cases (No = 1, No = 3, No = 4, and No = 6). No pain was elicited by deep palpation and manipulation of the thoracic limb in four cases (No = 4, No = 5, No = 7, and No = 8), unlike the remainder (No = 1, No = 2, No = 3, and No = 6).

Radiographs with special emphasis on the elbow and shoulder joints did not reveal any abnormalities in four cases (No = 1, No = 2, No = 5, and No = 6), while a decrease in joint space in the area of the contracture was shown in the other four (No = 3, No = 4, No = 7, and No = 8) (Figure 3 and Figure 4). All of the dogs underwent tenotomy of the infraspinatus muscle. In three of them, tenotomy of the teres minor muscle was deemed necessary during surgery, as it appeared thickened, with an increased diameter and under contracture leading to the reduced release of the shoulder joint’s motion (No = 1, No = 2, and No = 5). Partial tenotomy was performed in two of them (No = 1 and No = 5), with a total incision to its insertion in the other one (No = 2).

Postoperatively, strict limitation of activity for two weeks with a gradual return to physical activity was recommended and applied in most cases (No = 1, No = 2, No = 5, No = 6, No = 7, and No = 8). The dogs were postoperatively restricted to short walks on a leash combined with running and jumping limitations. In two cases, exercise restriction was applied for only one week (No = 1 and No = 4). The owners rated this measure as very helpful and important for their dogs’ mobility.

Medication was given in all cases, and the dog owners were satisfied with its efficacy. After 15 days, the animals presented for re-examination, where full recovery of the gait and a normal range of motion of the shoulder joint were noted. The outcome was excellent, as lameness was resolved in all cases over a period of 6 to 15 days; it was similarly rated as excellent by the owners, as all of them declared that they would choose the surgical treatment again and would also recommend it to another dog owner if needed. However, seroma appeared in two of the dogs (No = 1 and No = 7), which resolved in 1 week with warm compression. The time interval with toe-touching weight bearing for the limb undergoing tenotomy was from 2 to 4 days (mean = 2.75, median = 3) and was evaluated by the owners, while the range of the final weight bearing was from 6 to 15 days (mean = 10.75, median = 11) and was evaluated at the time of re-examination (Table 4).

According to the questionnaire answers, all the owners noticed that their dog’s quality of life was the same as before the surgery, while the lameness was rated as fully resolved, even after exercise. The activity level increased in two dogs (No = 2 and No = 3), remained stable in the rest of them, and the movement speed increased in one dog (No = 3). According to three owners (No = 2, No = 3, and No = 7), their dogs’ willingness to play increased. Friendly behavior toward people and other animals remained unchanged in all of them. Finally, the dog’s exercise endurance increased in three cases (No = 2, No = 3, and No = 6), and their physical condition improved in three cases (No = 1, No = 6, and No = 8).

## 4. Discussion

FCIM, also known as ‘the grey dog disease’ because of its high prevalence in the Norwegian Elkhound breed [19,30], is considered to be one of the two most frequent muscle contractures described in dogs [8,15]. Most cases involve working or hunting dogs that present with an acute onset of shoulder lameness with work or exercise, as also found in our study. Flexion of the shoulder joint results in external rotation of the humerus and abduction of the distal limb. Circumduction of the limb during the swing phase of the stride and lateral flipping of the paw are typical clinical signs [7,11,13,14,15,31], from which even our cases did not deviate. In most cases, the clinical histories do not reflect a specific traumatic episode, but it seems most likely that these dogs have been subjected to repetitive minor trauma, during exercise, hunting, or working [19]. This observation is also compatible with our study, as sudden lameness was the result of trauma in only three out of eight cases.

The dogs included in this study were affected by a suddenly occurring and persistent lameness that was unresponsive to NSAIDs. During the physical examination, the majority of dogs included were presented with shoulder muscle atrophy, while a reduced range of shoulder flexion and absence of shoulder pain was evident in the half of cases. All of the dogs underwent tenotomy of the infraspinatus muscle, while in three of them, tenotomy of the teres minor muscle was deemed necessary intra-operatively, as it appeared under contracture leading to the reduced release of the shoulder joint’s motion. Postoperatively, a strict limitation of activity for two weeks with a gradual return to physical activity was recommended and applied in most cases. Return to function was achieved within 15 days in all cases.

The initial pain and lameness seem to subside with exercise restriction and NSAIDs, but the characteristic gait abnormalities appear days to weeks later [15]. Our clinical and historical findings reinforce these bibliographic references. For diagnostic purposes, our patients primarily underwent clinical and radiological examinations. According to Butterworth and Cool (2006) [20], a reduction in the joint space between the greater tubercle and the rim of the glenoid cavity in the caudo-cranial radiographic view of the shoulder could support the diagnosis of infraspinatus muscle contracture. In addition, evidence of tendon mineralization in tangential radiographic views taken of the intertubercular groove of the shoulder joint could help in the diagnosis [15].

In our cases, the joint space reduction was evident in four cases. However, the second indication was not identified, maybe due to the subtlety and the susceptibility of these changes in joint space to postural changes during radiographic positioning. The crucial diagnostic tool is considered to be ultrasonography, in which the degenerative muscular changes can be revealed as disorganized hyperechogenic areas [7,8,31]. MRI is being used in the diagnostic workup of shoulder lameness [32,33,34], while Mikkelsen and Ottesen (2021) [22] support the use of CT imaging as an adjunct imaging modality for the evaluation of dogs with suspected muscle and/or tendon injury, through an FCIM case. Histopathology results could be very helpful both to the diagnosis and to the understanding of the pathology. The main findings are hyaline degeneration, active degeneration, necrosis, hyperemia, edema, and extensive hemorrhage of the infraspinatus muscle at the site of rupture [19]. Unfortunately, in our cases, they do not exist due to the refusal of the owners.

The exact cause of infraspinatus muscle contracture is not known; however, trauma appears to be most likely [15]. Pettit et al. (1978) [35], based on electrophysiological studies, suggested that the cause of FCIM is most likely myopathic rather than neuropathic. Bilateral involvement was mentioned in two reports [6,13], from which it arises that the injury may not be precipitated by a single violent event, but that it could be caused by overextension or overexercise [8]. More recently, it was suggested that the infraspinatus muscle in dogs is prone to developing compartment syndrome due to its confined osteofascial space [19,25].

Surgical treatment using a craniolateral approach and by performing a long tenectomy is the best choice in the chronic stage of this condition [8,21,34]. If compartment syndrome is present, a fasciotomy must be performed immediately succeeding surgical decompression to restore vascular perfusion, which should be based on the surgeon’s interpretation of the clinical signs [19,36]. Full recovery of gait and immediate return to function is expected in between 4 and 10 days [7,10,26], which agrees with our results too, with a deviation of five days. Postoperatively, restriction of activity for four weeks and physiotherapy (physically flexing and extending the shoulder four to six times daily) after suture removal is usually recommended [15,37]. Instead of physical therapy, we suggested exercise to prevent the development of fibrous adhesions between the infraspinatus tendon and the joint capsule, causing reoccurrence of the abnormal gait, but the exercise was in restricted form in order to avoid seroma formation [7]. We recommended walking the dog with a leash one-fourth of a mile three to four times daily after the second week postoperatively [10]. The absence of physical therapy does not seem to have affected the gait recovery in any of the cases, in terms of either time or completeness [19].

To our knowledge, this is the first published report describing a simultaneous tenotomy on both the teres minor and infraspinatus muscles in the same shoulder in three cases. In one case, tenotomy of the supraspinatus and infraspinatus muscles in the same limb of a dog was reported with an immediate return of function of the affected joint postoperatively [26,37]. Bruce et al. (1997) [28] reported a case of a teres minor myopathy in a working labrador retriever. In that case, the diagnosis was made based on the ultrasonographic and histopathological findings. The etiology of this condition is unknown, and we can only assume that strenuous activity and trauma were initiating factors based on the pathogeny of FCIM. According to the authors, no gross evidence of contracture or tears in the teres minor muscle was found. Treatment of this condition involved the removal of the entire teres minor muscle, resulting in only a slight increase in the external and internal rotation due to the rotator cuff muscles’ removal, without a detectable decrease in the shoulder joint stability.

In our cases, contracture of the teres minor muscle was revealed as preventing the full range of motion in the shoulder joint despite the tenotomy of the infraspinatus muscle. The teres minor muscle acting as a flexor caused the shoulder’s extension reduction [26]. These dogs did not present with clinical findings different from FCIM, which may mean that the clinical signs of this condition are obscured by the former. As a result, we conclude that in every case of shoulder lameness, contractures of the other shoulder muscles should be included in the differential diagnosis and checked during the surgery. In our study, this condition resolved with partial and total incision to the teres minor’s insertion close to the humeral bone, which immediately restored the normal motion of the joint.

In the context of the limited infraspinatus and teres minor contracture literature based on clinical cases, we consider that our study contributes to its enrichment by adding information about the epidemiology, case management, and outcome. However, one limitation of the present study is the lack of objective criteria for the one-year post-operative progress evaluation; consequently, incomplete or biased recall of events by the owners is possible due to the length of time that passed between the surgery and the completion of the questionnaire. Unfortunately, the reliability and validity of these answers cannot be confirmed. Therefore, this procedure makes the results subjective and acceptable with reservations, but their importance should not be diminished given the objective findings found during the re-examination.

## 5. Conclusions

In conclusion, FCIM is a rare condition but with a very typical clinical and epidemiological presentation that helps in its diagnosis. Its treatment is tenotomy of the tendon at its insertion, immediately restoring limb function. This paper presents three cases of simultaneous contracture of the teres minor muscle. These animals did not deviate from the clinical picture of FCIM. On the basis of these cases, the prognosis of this condition with a similar tenotomy seems to be excellent.

## Figures and Tables

**Figure 1 animals-14-02589-f001:**
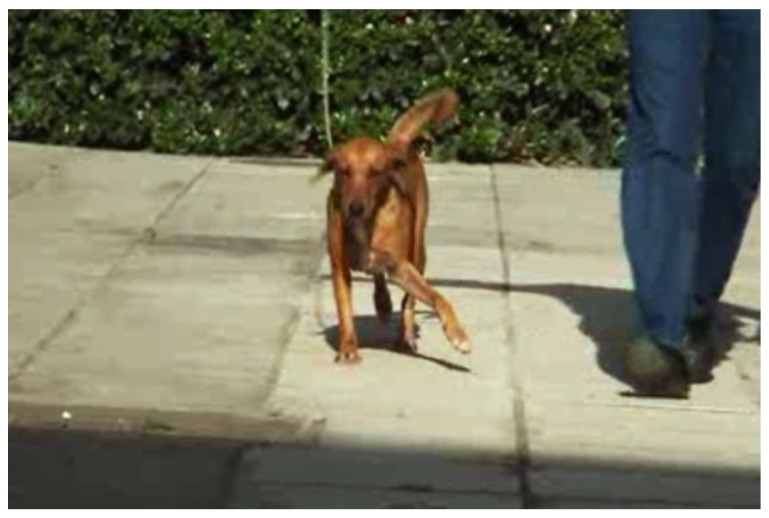
The distal part of the left thoracic limb swung in a lateral arc, drawing a circumduction movement as the foot advanced during the stride (No = 2).

**Figure 2 animals-14-02589-f002:**
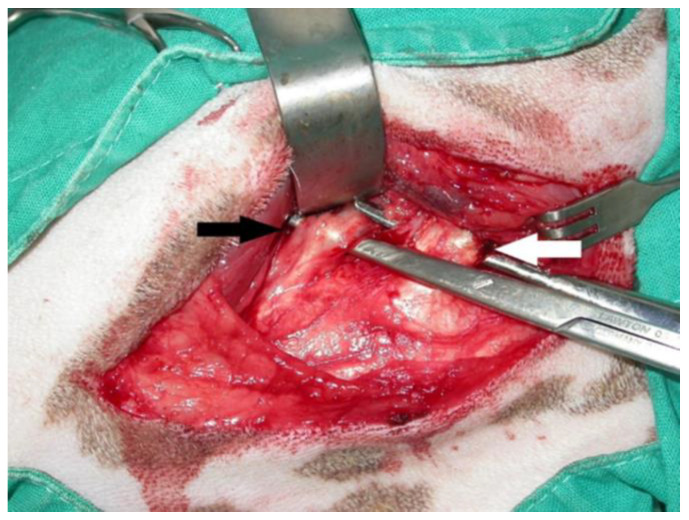
Intraoperative photo of infraspinatus and teres minor muscle contracture. White arrow: teres minor muscle. Black arrow: infraspinatus muscle. The dog is in left lateral recumbency.

**Figure 3 animals-14-02589-f003:**
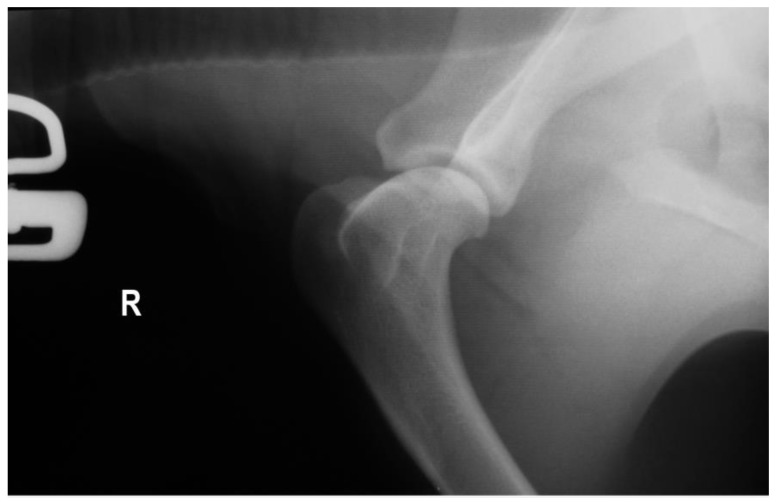
Preoperative mediolateral radiograph of the right shoulder (No. 6, Table 3) with infraspinatus m. contracture. No radiographic findings are shown.

**Figure 4 animals-14-02589-f004:**
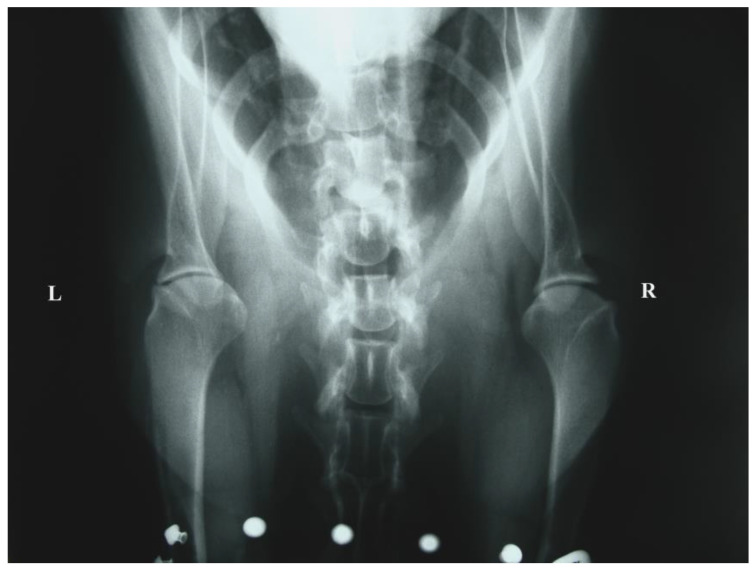
Preoperative craniocaudal view of both shoulder joints (No. 5, Table 3). Infraspinatus and teres minor m. contracture on the left, without radiographic signs.

**Table 1 animals-14-02589-t001:** Questionnaire.

A/A:………….……………				
Registration Number:…………………………………………				
Presentation Date: ……………… Surgery’s Date:…………….				
Owner: …………………………………………………………………				
Phone number: ………………………………………………………				
Dog’s Characteristics: Male □ Female □ Neutered □				
Age:……………..……………Breed:………………….…				
Name:……………..…………Weight:……………………				
Disease:				
Fibrotic contracture of the infraspinatus muscle □				
Fibrotic contracture of the teres minor muscle □				
Completed questionnaire: Yes □ No □				
1. Preoperative lameness				
1.1. Duration:……………………				
1.2. Grade: ⓪ ➀ ➁ ➂ ➃ ➄				
2. Postoperative lameness				
2.1. When began dog’s limb’s weight-bearing?				
2.2. When did the limb exhibit full weight bearing?				
2.3. Lameness grade one year postoperatively ⓪ ➀ ➁ ➂ ➃ ➄				
3. Postoperative analgesia				
3.1. Yes □ No □				
3.2. Drug:				
3.3. Administration’s duration:………………………….…				
3.4. Was analgesia useful? Yes □ No □ Possibly □				
4. Postoperative restriction				
4.1. Yes □ No □				
4.2. Duration:………………				
4.3. Restriction’s kind:…………………….				
4.4. Was restriction useful? Yes □ No □ Possibly □				
5. Does the dog appear lameness after exercise?				
5.1. Yes □ No □				
5.2. Grade: ⓪ ➀ ➁ ➂ ➃ ➄				
5.3. Does the lameness reduce after rest? Yes □ No □				
6. Assessment of dog’s postoperative clinical condition				
1. Worsening □				
2. Stable □				
3. Small improvement □				
4. Great improvement □				
5. Full recovery □				
7. Other questions				
7.1. Do you think that your dog’s quality of life is the same as before?				
……………………….…………………………………………………………………				
7.2. If you could decide again, would you make the same decision?				
…………………………………………………………………………………………				
7.3. Would you recommend this treatment to someone you know for his dog?				
…………………………………………………………………………………………				
8. Differences in dog behavior/activity after surgery				
*Behavior/Activity*	*Reduction*	*Stable*	*Increase*	*I don’t know*
8.1. Activity grade				
8.2. Movement speed				
8.3. Mood for playing				
8.4. Physical condition				
8.5. Friendly attitude toward people				
8.6. Friendly attitude toward other animals				
8.7. Endurance				

**Table 2 animals-14-02589-t002:** Lameness scale (Krystalli et al. 2023) [29].

Degree of Lameness	Limb’s Weight Bearing				Characterization of Lameness
	Description	Stance	Walk	Trot	
0	Full (normal) weight bearing	......	......	......	Absence
1	Partial weight bearing: hardly visible	......	......	......	Light
2	Partial weight bearing: easily visible	......	......	......	Mild
3	No weight bearing: intermittent, sporadic (≤1:5) *	......	......	......	Moderate
4	No weight bearing: intermittent, frequent (>1:5) *	......	......	......	Severe
5	No weight bearing: continuous	......	......	......	Not functional
Degree of lameness = (S + W + T)/3					

*: limb lift frequency per 5 steps.

**Table 3 animals-14-02589-t003:** Summary of treated animals.

Dog’s Number (n)							
	Breed	Age (Years)	Gender	Body Weight (kg)	Thoracic Limb	Grade of Lameness	Duration of Lameness (Days)
1	Pointer	4	Male	26	R	2	90
2	Balkan hound	5	Female	17	L	2	60
3	English Setter	7	Female	17.5	R	1	45
4	Mongrel	9	Male	14.2	R	1	60
5	Greek Harehound	5	Female	21.5	L	2	60
6	Mongrel	9	Male	25.5	R	1	450
7	English Setter	6	Male	18	R	1	10
8	Mongrel	5	Female	14	L	2	120

R-right, L-left.

**Table 4 animals-14-02589-t004:** Summary of the limb’s time initial and final weight bearing.

Dog’s Number (n)	Time of Initial Weight Bearing	Time of Final Weight Bearing
1	3	6
2	2	10
3	2	14
4	3	15
5	2	8
6	3	6
7	4	15
8	3	12

## Data Availability

The original contributions presented in the study are included in the article. Further inquiries can be directed to the corresponding author.

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
