# Peer review of "Fibrotic Contracture of the Infraspinatus Muscle with or without Contracture of the Teres Minor Muscle: A Retrospective Study in Eight Dogs"

_animals, 2024, doi:10.3390/ani14172589_

Round 1

Reviewer 1 Report

Comments and Suggestions for Authors

The paper “Fibrotic contracture of the infraspinatus muscle with or without
contracture of the teres minor muscle: Α retrospective study in eight dogs” is an interesting topic but needs improvement

Introduction

The introduction contains phrases without author references, which is inappropriate. Please ensure proper referencing for all statements.

The introduction includes a case illustration, which should be moved to the Materials and Methods section if it is your case.

If the case is not yours, permission is required to use it.

Materials and Methods

Table 1 should be included as a supplementary file.

The study has two significant limitations: the lack of imaging exams other than X-rays to confirm muscle lesions and the absence of histological examination. These limitations should be highlighted.

Results

Since the diagnosis lacks support from imaging or histological exams, illustrations of cases before and after the treatment should be included. If you have movies, they would be of interest.

Discussion

The discussion section needs to be rewritten.

The first paragraph of the discussion is more of a literature review than a discussion. Focus on presenting your findings and discussing them concerning to existing literature.

As observed in the introduction, you use phrases from other authors but the reference is not provided.

Specific comments have been provided in the PDF.

Author Response

Thank you for your comments. Below is our answer.

Is the English used correct and readable?  

Response: We used the Animals editing services to make English of our study correct and readable.

The introduction contains phrases without author references, which is inappropriate. Please ensure proper referencing for all statements.

Response: We referenced all statements. Nevertheless, we have added some according to your suggestions

The introduction includes a case illustration, which should be moved to the Materials and Methods section if it is your case.

Response: The case is ours. We moved it to the Materials and Methods

Table 1 should be included as a supplementary file.

Response: We have no doubt. We placed it in the main text because at Τhe instructions for authors it is stated: "All Figures, Schemes and Tables should be inserted into the main text close to their first citation and must be numbered following their number of appearance (Figure 1, Scheme 1, Figure 2, Scheme 2, Table 1, etc.)."

The study has two significant limitations: the lack of imaging exams other than X-rays to confirm muscle lesions and the absence of histological examination. These limitations should be highlighted.

Response: We have pre-operative x-rays and we added two of them. Unfortunetely according the post-operative they don't exist, as when the owners see improvment in their animals' movement, they don't easily accept to anaisthesiaze their animals again for x-rays.

According histological examinations it would be very interesting to have but unfortunetely the owners did not agree.

Since the diagnosis lacks support from imaging or histological exams, illustrations of cases before and after the treatment should be included. If you have movies, they would be of interest.

Response: We add a video from one of our cases.

The discussion section needs to be rewritten.

Response: We tried to follow your next comment and we think it improved the discussion

The first paragraph of the discussion is more of a literature review than a discussion. Focus on presenting your findings and discussing them concerning to existing literature.

Response: We added a paragraph with our findings.

As observed in the introduction, you use phrases from other authors but the reference is not provided

Response: We referenced all statements.

Specific comments have been answered in the attached PDF.

We also add two videos from one case. Let us know if you want to publish them

Reviewer 2 Report

Comments and Suggestions for Authors

The authors describe a restrospective study on fibrotic contracture of the infraspinatus muscle. The study has been carried out on 8 dogs, an apparently very low number, but congruent for a clinical point of view. 

The study appears well conducted. The introduction is broad and quite complete with numerous references. Material and methods and results are convincing and the discussion seems appropiate. 

I think that this manuscript deserves to be taken in consideration for publication after two minor points.

1) when the tenotomy is decribed, the authors shoud specify which tendon is being sectioned:  prossimal of distal?

2) the authors should include at least one X-ray of the scapular-humeral joint (preferably pre and post tenotomy).

Author Response

Thank you very much for your comments and for reviewing our manuscript. Below is our answer to your comments:

1) when the tenotomy is decribed, the authors shoud specify which tendon is being sectioned:  prossimal of distal?

Response: Line 184 “incision 1 cm close to its insertion to the greater tubercle”

2) the authors should include at least one X-ray of the scapular-humeral joint (preferably pre and post tenotomy).

Response: We have pre-operative x-rays and we added two of them. Unfortunetely according the post-operative they don't exist, as when the owners see improvment in their animals' movement, they don't easily accept to anaisthesiaze their animals again for x-rays.

Reviewer 3 Report

Comments and Suggestions for Authors

Reviewer comments for manuscript ID animals-3143781 entitled ’Fibrotic contracture of the infraspinatus muscle with or without contracture of the teres minor muscle: Α retrospective study in  eight dogs’

General Comments

Musculosketetal injuries in working an hunting dogs present diagnostic as well as therapeutic challenges to clinicians. Injuries in the forelimbs are more crippling owing to the maximum weight bearing on these pair of limbs. Fibrotic contracture of the canine  infraspinatus muscle (FCIM) along with the involvement of teres mnor muscle requires high level of diagnostic competence at the level of clinicians as incidental finding of the involvement of the latter muscle during surgery itself presents a challenge as well as a risk on the outcome of surgery.

 The authors have presented a commendable work that will be useful for small animal orthopaedicians. The manuscript is nicely written and well presented with tables and figures. Introduction is to the point and creates the interest of the reader. Materials and methods and the results logically lead to a well written discussion with highlighting of the limitations of the study. I have few queries that I have pointed out specifically that pertains to the selection bias in the study and objectivity of the post operative review of the cases. I would also like to see the Ultrasonography images of the condition that will create more interest to the reader.

 I would like to see the answers to my queries before I recommend the publication of the manuscript.

Specific Comments

Line 165: ‘Good general health condition during surgery.’ Please clarify why this was one of the inclusion criterion. Does it introduce selection bias in the study?

Line 194-200: Examination of patient for evaluation of lameness is important for clinical assessment of recovery. Do you think owners’ assessment can differ than that of a clinician? How was this observer bias eliminated in the current study ? Please clarify.

Author Response

Thank you very much for your comments and for reviewing our manuscript. According your specific comments, our answer is the following:

I would also like to see the Ultrasonography images of the condition that will create more interest to the reader.

Unfortunetely we don't have any ultrasonography images due to owners refusal. Since we had the diagnosis they refused to pay an extra examination.

Line 165: ‘Good general health condition during surgery.’ Please clarify why this was one of the inclusion criterion. Does it introduce selection bias in the study?

In this study for the first time it is described the time of initial and final weight bearing. We didn't want any other factor to affect these two parameters.

Line 194-200: Examination of patient for evaluation of lameness is important for clinical assessment of recovery. Do you think owners’ assessment can differ than that of a clinician? How was this observer bias eliminated in the current study ? Please clarify.

Response: Τhe time of initial weight bearing is easily evaluated by the owners, so we believe that it is an objective criterion and this is the reason for its use in this study. The time of final weight bearing was evaluated by us.

Reviewer 4 Report

Comments and Suggestions for Authors

Thank you for your manuscript on infraspinatus contracture with several cases of concurrent teres minor contracture. It is an interesting paper and I found it to be well written and easy to understand and follow. I have just a few suggestions: 

Line 37-38: The concluding statement of the abstract is a bit awkward to read. I think tweaking the final portion of the sentence “...checked during the surgery...” to say that patients should be assessed for concurrent contractures when infraspinatus contracture is identified.  

Line 155: Typo, “gate” should be “gait” 

Table 2: Can you please clarify if “run” is being used to describe trotting, cantering, or galloping?  

Line 310: Missing a period between these sentences “...of five days. Postoperatively, restriction...” 

Line 320: Shouldn’t this say three cases?

Author Response

Thank you very much for your comments and for reviewing our manuscript. According your suggestions:

Line 37-38: The concluding statement of the abstract is a bit awkward to read. I think tweaking the final portion of the sentence “...checked during the surgery...” to say that patients should be assessed for concurrent contractures when infraspinatus contracture is identified.  

We have changed it.

Line 155: Typo, “gate” should be “gait” 

Corrected.

Table 2: Can you please clarify if “run” is being used to describe trotting, cantering, or galloping?

Trotting. We added it.

Line 310: Missing a period between these sentences “...of five days. Postoperatively, restriction...” 

Corrected.

Line 320: Shouldn’t this say three cases? 

You are right. Also corrected.

Round 2

Reviewer 1 Report

Comments and Suggestions for Authors

The article has improved significantly with the changes. The radiographic image in Figure 4 should be enhanced in quality by removing the green color. Furthermore, the description provided is inaccurate as the term "anteroposterior view" is not suitable for an animal.

Author Response

The article has improved significantly with the changes. The radiographic image in Figure 4 should be enhanced in quality by removing the green color. Furthermore, the description provided is inaccurate as the term "anteroposterior view" is not suitable for an animal.

Response: Thank you again for your comments. We removed the green colour from figure 4 and replaced the "anteroposterior" with craniocaudal